# Microbial Colonization of Explants after Osteosynthesis in Small Animals: Incidence and Influencing Factors

**DOI:** 10.3390/vetsci11050221

**Published:** 2024-05-16

**Authors:** Mario Candela Andrade, Pavel Slunsky, Tanja Pagel, Ignacio De Rus Aznar, Mathias Brunnberg, Leo Brunnberg

**Affiliations:** 1Department of Medicine, Health and Medical University, 14471 Potsdam, Germany; 2Small Animal Hospital, Anicura Kleintierspezialisten Augsburg, 86157 Augsburg, Germany; pavel.slunsky@anicura.de; 3Small Animal Clinic, Freie Universitaet Berlin, 14163 Berlin, Germany; tanja.pagel@fu-berlin.de (T.P.); leo.brunnberg@fu-berlin.de (L.B.); 4Shoulder Surgery Unit, Orthoapedic and Traumatology Department, CEMTRO Clinic, 28003 Madrid, Spain; ignacio.derus@clinicacemtro.com; 5Small Animal Clinic, Tierarztpraxis Sörensen, 12207 Berlin, Germany; mathias@brunnberg.de

**Keywords:** infection, hardware, implants, dogs, cats

## Abstract

**Simple Summary:**

Despite medical advancements, post-operative infections in osteosynthesis remain common, causing complications like delayed healing and implant failure. Investigating microbial colonization after surgery in small animals, this study aims to understand infection rates and influencing factors. Results from 71 explants show correlations between infection, patient characteristics, and surgical variables. Notably, factors like body weight, implant type, and additional injuries impact infection development. Surprisingly, while microorganisms were present in nearly half of the cases, only 7.3% led to clinical complications. These findings highlight the complexity of post-operative infections and suggest that microbial presence does not always lead to complications, offering insights for improved treatment strategies in veterinary medicine, potentially reducing patient suffering and healthcare costs.

**Abstract:**

Despite recent advancements in antibiotics, hygienic measures, and peri-operative systemic antibiotics, post-operative infections in osteosynthesis remain prevalent and continue to be among the most common surgical complications, leading to delayed fracture healing, osteomyelitis, implant loosening, and loss of function. Osteosynthesis implants are routinely utilized in veterinary medicine and the current study investigates the microbial colonization of implants following osteosynthesis in small animals, along with its incidence and influencing factors. The results are analyzed in regard to correlations between infection, patient, disease progression, and radiographic images, as well as other factors that may promote infection. Seventy-one explants from sixty-five patients were examined and evaluated for microbial colonization. Factors like body weight and age, location and type of plate and additional injuries like lung lesions, the surgeon’s experience, or the number of people present during the surgical procedure seem to influence the development of an infection. Of the animals, 60% showed osteolytic changes and 73.3% of those with dysfunctional mobility had an implant infection. Microorganisms were detected in almost 50% of the explants, but a clinically relevant infection was only present in five patients (7.3%), suggesting that the presence of microorganisms on an implant does not necessarily lead to treatment complications.

## 1. Introduction

One of the most important tasks of orthopedics is the restoration of the function of body parts with the help of orthopedic implants. Overall, infectious complications of elective orthopedic surgeries are rare with an average infection rate of 5% of the osteosynthetic materials used in human medicine [1]. The incidence of infection in closed fractures is usually low (1–2%), while in open fractures it can be up to 30% [2]. In human medicine, post-operative infections still count, even when considering the recent development of antibiotics and peri-operative systemic antibiotics, as the most common surgical complications and the consequences include delayed fracture healing, osteomyelitis, implant loosening, and loss of function [3,4]. Khan et al. [5] pointed out that infection occurred in 6 of 104 patients after osteosynthesis.

The pathogenesis of many infections in surgery and the development of microorganisms in biofilms are influenced by different factors [6]. Especially if foreign bodies are implanted, a biofilm may develop on the body’s surface and this is especially dangerous if these remain in place permanently. Biofilms are a significant problem in treating bacterial infections and are one of the main reasons for the persistence of infections [7]. Considering that implants are commonly employed in veterinary medicine and bacterial infections have the potential to impact fracture healing, the significance of this present study cannot be overstated. It delves into the microbial colonization of implants post-osteosynthesis in small animals, analyzing its occurrence and the factors that influence it. The findings of this research hold immense importance in enhancing our understanding of post-operative complications and optimizing treatment strategies in veterinary orthopedics.

## 2. Materials and Methods

The aim of this work was to examine explanted osteosynthesis plates for bacterial colonization. The results are analyzed in regard to correlations between infection, patient, disease progression, and radiographic images, as well as identifying any pathogenetic factors that may promote infection.

### 2.1. Patients

The study included patients presented between February 2010 and March 2013 at the small animal teaching hospital of the Freie Universität Berlin (Berlin, Germany) for an explantation procedure.

The recorded data included: breed, age, gender and weight, type of injury, location, additional injuries, time between accident and surgery, duration of the hospitalization period, previous surgeries, antibiotic treatment, type of plate used for the osteosynthesis, additional implants, surgeon and team, duration of the surgical procedure, time between surgery and explantation, radiographic findings before implant removal, and complications after implant removal.

Microbiological findings from the plates were sent to the Institute of Microbiology and Animal Diseases of the Freie Universität Berlin (Berlin, Germany) for analysis.

To determine the type of injury and location, radiographic images of traumatized limbs in two perpendicular planes were used. In case of any joint involvement, additional stress radiographs were also taken to determine the extent of the dislocation and the exact location of the injury and to identify proximal, distal, epiphyseal, or metaphyseal lesions.

Patients presented to the clinic were protocolarily examined in order to diagnose additional injuries like pneumothorax, other wounds, or abdominal bleeding with thoracic and abdominal radiographic images. Filed data included not only radiographic images but also sonographic or computed tomographic examinations.

The time between accident and surgical intervention in the clinic, as well as any previous interventions and whether antibiotics were used pre-, peri-, or post-operatively over a longer period of time, was registered and analyzed to deduce any influence on the healing process. Further information like previous surgical interventions, especially in connection with the current issue, as well as other non-surgical diseases, especially potential sources of infection such as pyoderma, were included in the documentation analyses. This also included documentation for any antibiotic treatments, alio loco.

Osteosynthesis plates and any additional implants, such as cerclages, screws, Kirschner wires, the plate type (DCP, locking plates, T-plate) or its thickness (2.0, 2.7, 3.5, and 4.5 mm), and the number of plate holes were also documented. The surgical team, the duration of the procedure, and the hospitalization, with a focus on the surgeon and assistants as well as their numbers, were analyzed. Surgeons were classified as highly qualified (diplomate of the European College of Veterinary Surgery and/or specialist veterinarian for surgery) or not yet qualified (resident, training to become a specialist veterinarian). The number of additional people in the room during the procedure or for how long they were present could not always be reliably processed since some documentation was missing. The duration of the intervention and the total time of hospitalization were also included in the analysis.

Early post-operative complications and time between surgery, implant removal, and radiographic findings, such as post-operative disorders of wound healing in the form of infection, fistula formation, or implant failure (loosening, breakage), were documented. In patients that underwent surgery for a second time due to complications, radiographic images before implant removal in those scenarios (implant loosening, implant bending or fracture, osteolysis, sequestration, pseudoarthrosis, bone demineralization, or refracturing of a bine) as well as complications that arose during surgical explantation were analyzed. During implant removal, swabs for microbiological analysis were taken and transferred to the Institute of Microbiology and Animal Diseases of the Freie Universität Berlin (Berlin, Germany).

### 2.2. Implants

Plates, screws, and additional implants removed in this study were products from Königsee Implants (Hamburg, Germany). Locking plates and dynamic compression plates (DCPs) with a thickness varying from 2.0 to 4.4 mm and 6 to 16 holes were analyzed to check for possible bacterial colonization. In addition, 8-hole 2 mm T-plates were implanted, removed, and analyzed for bacterial colonization.

The removal of the implants followed the standardized approach to each bone described by Piermattei and Flo [8]. Once the implant site was reached, the connective tissue surrounding the plate was cranially or caudally dissected along the base of the plate near the bone. Following this, the connective tissue was folded so that the screw heads were exposed and freed of ingrown connective tissue.

### 2.3. Microbiological Examination

Explants were sampled for microbiological examination in two ways:

Sterile microbiological swabs were taken intraoperatively when the explant was exposed but not yet removed. Sterile swabs from Heinz Herenz (Hamburg, Germany) were immediately placed in the sterile medium within the swab tube and sent to the Institute of Microbiology and Animal Diseases of the Freie Universität Berlin (Berlin, Germany).Dilution smears were made directly from the implant on various culture media and bouillons. For each implant, 5 nutrient agars and 2 nutrient broths were used, all of which were provided by the same institute. These included 3 aerobic culture media (chocolate agar, Columbia agar, urine chromogenic agar), 2 anaerobic culture media (Columbia agar anaerobic, gentamicin agar), and test tubes with an aerobic and anaerobic brain–heart infusion broth (BHI-bouillon).

After removing the plate with the screws and isolating it from the rest of the connective tissue, the material was placed on a prepared sterile surgical table for further processing. Two swabs were used to take samples from the removed implant, and a swab was placed on the culture medium to then create fractional dilution smears using the inoculation loops. Both swab tips were then cut off with sterile Cooper scissors, and one of each nutrient broth was inserted for incubation. All anaerobic nutrient media were sent in sealed Zeissler pots after adding the AnaeroGen gas (Oxioid, Wesler, Germany) for further analysis.

### 2.4. Statistical Analysis

The data were analyzed using SPSS (v22, IBM, Armonk, NY, USA, 2020). The significance was determined as *p* < 0.05. Non-parametric tests were used to calculate the significance. The chi-square test, Fisher’s exact test, and Mann–Whitney U test were used to determine any significant univariable associations between the development of bacterial colonization on an explant and its bacteriological results with other factors like breed, weight, sex, age, indication for explantation, fracture location and concomitant lesions, antibiotic therapy, time until surgery, previous surgical procedures, type of plates and additional implants used, number of assistants and surgeons, duration of the surgical procedure and stay at the clinic, time until explantation, as well as complications during the healing phase or after explantation and radiographic findings before explantation.

## 3. Results

In this study, 65 cases of patients presented between February 2010 and March 2013 for an explantation procedure at the Small Animal Clinic of the Freie Universität Berlin (Berlin, Germany) were analyzed. Fifty-one of the animals were dogs and fourteen were cats. In the case of six dogs, two plates with screws were removed. In three of these six patients, implants were placed and removed on the same day. In the case of the other three dogs, the implants were implanted and removed on different occasions. A total of 71 plate implants from 65 patients explanted in 68 operations were examined and evaluated.

### 3.1. Breed, Age, Sex, and Weight

Figure 1 shows the different dog breeds included in the study. Amongst the fourteen cats were one Norwegian Forest Cat, one Abyssinian, one Persian, and ten European Shorthairs. From a total of 51 dogs, 29.4% (*n* = 15) were mixed breeds.

The 14 cats were between 0.7 and 12 years old, with a mean of 3.9 and a median age of 2.5 years. Dogs (*n* = 51) were between 0.4 and 14 years old. The mean age was 4.0, with a median of 3 years. Thirty-seven dogs were young, between 0 and 5 years old, and fourteen were older, between 5.1 and 14 years old. There was no statistically significant correlation between age and explant infection (*p* = 0.511).

From a total of 14 cats, 2 were intact males, 8 were neutered, 2 were intact females, and 2 were spayed. Of the 51 dogs, 18 were intact males, 9 were neutered, 18 were intact females, and 6 were spayed females. No statistical correlation between sex and explant infection was found (*p* = 0.726).

The body weight of the cats (*n* = 14) varied between 3 kg and 8.5 kg. The mean weight was 4.7 kg and the median weight was 4.2 kg. Cats were classified into weight classes. One weighed less than 3 kg, the majority (*n* = 10) between 3.1 and 5 kg, one cat weighed 5.1–7 kg, and two were heavier than 7 kg. The dogs’ weight varied between 1.9 and 55 kg, with a mean weight of 23.6 kg and a median weight of 24.2 kg (Figure 2).

A statistically significant correlation was found between heavier dogs and an explant infection (*p* < 0.05) (Table 1).

### 3.2. Indication for Surgical Intervention

Arthrodesis with plates and screws was performed in 14 dogs, with the carpal joint (*n* = 5) being the most commonly treated joint. Three of these patients were treated with two plates and two of them received only one. In two cases, arthrodesis was performed in the shoulder joint due to dysplasia and arthrosis. In addition, two knee and elbow joints and three tarsocrural joints were treated with an arthrodesis.

Both knee joints had previously been treated in a different facility several times because of a cranial cruciate ligament rupture and referred for further treatment due to an extremely painful ankylosis. One elbow was arthrodesed due to severe osteoarthritis. The other elbow arthrodesis was performed in a dog, in which the elbow joint was stiffened after it had already been operated on twice unsuccessfully after a complicated Y-T humeral condyle fracture. The arthrodesis of the tarsal joint was performed because of severe osteoarthritis. This dog had to undergo surgery twice because the first plate broke after surgery. In two other patients, the indication for arthrodesis of the tarsal joint was not included in the patient records. A corrective osteotomy was performed on four animals. Three of them had their tibias osteotomized. One dog suffered from tibial dysplasia, for the other two, the indication could not be found in the documents, and the fourth dog had a bilateral carpus valgus after an old trauma and had two surgeries performed in the left forelimb. As a result of a car accident, two dogs had a rupture of the long medial tarsal collateral ligament that needed to be surgically treated.

Thirty-one dogs (N = 51, 60.8%) were diagnosed with traumatic fractures and surgically treated. Three fractures were grade I open fractures. In 61.3% (n = 19) of dogs with fractures, the cause could be determined and the most frequent cause of fracture was a car accident in 73.7% of the cases, followed by bite injuries in 10.5% of the cases.

Two dogs originally underwent a total endoprosthesis as a result of hip dysplasia. In one of these two patients, the prothesis loosened, so another surgical procedure was necessary which led to a fracture of the femur intraoperatively while attempting to remove the shaft component of the prosthesis. The second dog also showed a dislocated and eventually loosened implant, but in this patient the femur fracture occurred not intra- but post-operatively. Both femur fractures were reported in this study because osteosynthesis was performed with plates and screws. The cats (N = 14) in this study were presented due to a fracture. Two cats had an open fracture (grade 1). In 71.4% of the cases, the cause of the fracture was known, and the most common cause was a fall from a height (64.3%; n = 9).

### 3.3. Localization of the Injury

In the present study, investigations were carried out on 65 dogs and cats, in which fractures of the forearm (n = 19) and carpus (n = 6) were more common than femoral fractures (n = 12), lower limb (n= 12), and tarsal fractures (n = 5). Four humeral fractures occurred, as well as two shoulder, two elbow, two knee joint, and one ileal fracture (Table 2).

The explants of the forelimb were infected more often, with n = 21 (60%), in contrast to those of the hind limbs, with n = 13 (39.4%). Interestingly, for the fore- and hindlimbs, the structures distal of the elbow (n = 18/21, 85.7%) and knee joint (n = 10/13, 76.9%), respectively, were more often infected than the more proximal structures.

### 3.4. Antibiotic Treatment

Apart from pre-operative antibiotics, further post-operative antibiotic treatment was administered after 30 operations (45.5%, N = 66). In two animals that were initially presented to a different facility, further post-operative antibiotic treatment was not found in the files. In 77.4% of patients, amoxicillin and clavulanic acid were used. Cefalexin, marbofloxacine, enrofloxacine, and trimethoprim–sulfonamide were used in 9.7%, 6.5%, 3.2%, and 3.2% of the cases, respectively. Only one animal had two antibiotics prescribed post-operatively.

Of these 30 patients treated post-operatively with antibiotic treatment, 17 (56.7%) had infected explants. Regarding the risk of infection between patients treated with and without antibiotic treatment post-operatively, no significant difference was found (*p* = 0.296).

### 3.5. Additional Injuries after Trauma

Nineteen dogs were not presented for a recent trauma and were therefore excluded in the following analysis. According to the patient records, 14 of the 32 dogs analyzed (43.8%) had further injuries after their traumatic event. Three dogs had more than one additional injury. In 50% of the cases, the most common diagnosed injury was another fracture (Figure 3). Ten of the fourteen cats (71.4%) had additional injuries, with two of the animals having more than one. Fifty percent of cats showed lung lesions which was the most commonly diagnosed comorbidity (Figure 1) and the only type of additional injury that was significantly correlated to infected explants (*p* = 0.028).

### 3.6. Time between Diagnosis of the Lesion and Surgical Treatment

The medical records of 40 patients (N = 65) revealed the period between the diagnoses of the disease or lesion and surgical treatment (Figure 4). In 25 cases, this was not possible because there were no anamnestic data. The risk of infection did not increase significantly with an increasing period of time between diagnosis/occurrence of the lesion and time of surgery (*p* = 0.915). 

### 3.7. Number of Previous Operations

Fifteen animals (23.1%) had already been surgically treated alio loco once (n = 7), twice (n = 4), three (n = 3), or four times (n = 1).

### 3.8. Type of Plate Osteosynthesis

During implant removal in the clinic, 37 locking plates, 28 dynamic compression plates (DCPs), and 6 T-plates were explanted. The number of screw holes in locking plates and DCPs and their thickness are compiled in Table 3 and Table 4. The T-plates were 2 mm thick with a total of eight holes.

Of the 28 explanted DCPs, 11 (39.3%) were infected. In contrast to this, 22 of 37 locking plates (59.5%) and 2 of 6 T-plates (33.3%) were infected. Despite the numerical difference, no statistical significance could be found for any of these plate types (*p* = 0.296).

In percentage terms, thin plates (2–2.7 mm) had a lower risk of infection (38.5%) than thicker ones (3.5–4.5 mm) (62.5%). Yet, no statistically significant correlation could be found between the thickness of the plate and the risk of infection (*p* = 0.328).

### 3.9. Additional Implant Material

The aim of plate osteosynthesis was to achieve load stability. This was achieved with plates and not always screws, especially in comminuted fractures, which was the case in 28 patients. While 19 patients required the addition of one more implant, 9 patients were in need of two additional implants. Wire cerclages (29.7%) were the most common additional implant, while screws (27%), Kirschner drill wires (16.2%), pins (16.2%), and plates (10.8%) were less commonly used for this purpose. When removing the material, half of these animals (n = 14) had a microbiological infection on the explants. In patients without additional osteosynthesis material, the percentage was 47.5%. No statistical significance was found between additional implant material and infection (*p* = 0.513).

### 3.10. Surgeon and Assistants

A total of 68 explantations were carried out in the clinic by seven different surgeons or the referring vet. The referring vet´s qualifications were not known. The surgeon’s experience was classified from inexperienced to highly qualified with years of experience (i.e., professors or diplomates) in osteosynthesis. The first two surgeons were highly qualified, experienced surgeons, while the five other operators were trained but still inexperienced. Forty-seven procedures (N = 68) were carried out by one of the highly qualified and experienced surgeons and eighteen by one of the five other surgeons from the small animal clinic. Three explantations were carried out by the referring vet. Thirteen operations were performed with one assistant, thirty-six operations with two assistants, and sixteen operations with a total of three people assisting. Three operations were carried out by the referring veterinarian, meaning the number of assistants remains unknown. Fourteen of the sixteen operations in which three assistants were present were carried out by one of the experienced surgeons. The different surgeons performed between 1 and 45 operations. The infection rate among experienced surgeons (nr 1 and 2) was 52% (n = 26/50) and among the inexperienced (nr 3 to 8) it was 42.9% (9/21) in relation to the total examined explants (N = 71). No statistically significant difference of the risk of infection between the experienced and inexperienced surgeons was proven (*p* = 0.582), but the most frequently infected explants (12/18, 66.7%) were those where three assistants were present during surgery.

### 3.11. Duration of Surgeries and Length of Hospitalization

Only 32 operations (47.1%) had the duration of the procedure documented in the anesthetic protocol. The operation times ranged from 30 to 180 min, with an average of 87 min.

Operations that lasted between 30 and 80 min (n = 15) in this study had an infection risk of 73.3% while interventions between 90 and 180 min (n = 17) had an infection rate of 47.1%. Explants that tended to take less time when implanted showed infectious pathogens more frequently than in those longer implantation procedures.

The length of hospitalization for 44 patients varied between 1 day and 13 days. The average hospital stay was 3.1 days. Twenty-one animals had an ambulant surgical treatment in the clinic and three at their referring vet (n = 24). As shown in Table 5, the risk of infection tended to decrease with the duration of hospitalization, yet no significant correlation was found between hospitalized and ambulant patients regarding the rate of implant infection (*p* = 0.802). Furthermore, no statistically significant correlation was found between the length of stay and the risk of developing an implant infection (*p* = 0.563).

### 3.12. Complications during the Healing Phase and Time to Implant Removal

In eight dogs, infection occurred in the post-operative healing phase, between initial treatment and implant removal, either in the form of a wound infection (n = 5), suture dehiscence (n = 1), or fistulation (n = 2). During the healing phase, nine dogs and three cats were operated on again to remove or replace parts of the implants to dynamize the fracture (n = 8), or both measures were carried out in one session, e.g., cerclage removal and screw change. In three patients, additional healing stimulation with a cancellous bone transplant was needed. In the animal with suture dehiscence, the wound was revised.

The time to removal of the osteosynthesis plates (n = 70) varied from 14 to 1658 days, with an average of 153.6 days. In one patient, the fracture was surgically treated by the referring veterinarian and it was not possible to determine the exact time of implantation. Of implants that remained in situ between 0 and 120 days, 43.2% were infected, while those that were removed at a later point in time were infected in 57.6% of cases. However, there was no statistically significant correlation between the rate of infection and the time to implant removal. In seven of eight patients (87.5%) who suffered a wound infection, fistula formation, or suture dehiscence post-operatively during the first healing phase, the explants were infected. Of the 12 patients that were operated on again due to wound healing deficits, infections were found on the explants of 8 patients (66.7%).

### 3.13. Radiographic Findings before Implant Removal

Thirty-six radiographic images (N = 68) had no special findings before implant removal. In 32 images, findings included osteolysis around the implant area (n = 19, 27.9%), bone demineralization under the plate (n = 9, 13.2%), broken implants (4× plates, 3× screws) (n = 7, 10.3%), loosened screws (n = 6, 8.8%), a bent plate (2.9%), sequestration (2.9%), or non-union (1.5%). No significant difference in terms of risk of infection was found in patients with or without radiographical changes (*p* = 0.627).

### 3.14. Indication for Implant Removal

The first indication for implant removal was that the fracture, arthrodesis, or corrective osteotomy had healed without complications. This was the case for 72.1% of patients (n = 49). There was no information regarding this aspect in the medical records of four patients. The implants were removed from 15 animals because of a high (n = 3), moderate (n = 8), or mild (n = 1) implant-related lameness, massive soft tissue swelling and limb misalignment (n = 1), or fracture instability (n = 1).

In 20 patients (40,8%) who had the implant(s) routinely removed after uncomplicated healing, the foreign material was infected. The infection rate was higher (73.3%, n = 11) in those who had a functional movement disorder, a limb deformity after healing, or inflammation.

However, there was no statistical significance for any risk of infection to be derived from a moderate (*p* = 0.144) or severe (*p* = 0.608) lameness.

### 3.15. Complications after Implant Removal

After implant removal, wound infections within the first 14 days (n = 4), refractures (n = 3), and non-stable fractures (n = 2) were diagnosed. The fractures were immediately treated with a new osteosynthesis.

The explant was infected in the four patients who suffered from wound infection after removal (*p* = 0.05). In two of the three patients suffering a refracture, an infection could be detected on the explant (*p* = 0.608). In contrast, the explants of both animals with fracture instability were free of infection after removal (*p* = 0.493). Clinically and radiographically, in 5 of 51 dogs (9.8%) osteomyelitis was diagnosed. These animals were between 0.5 and 11 years old and weighed between 22 and 53 kg. In two patients, only the tibia was affected; the other three patients had humerus, tibia, fibula, radius, or ulna fractures, respectively. Three patients were hospitalized for a day or more and developed a wound infection. In each case, the wound had to be surgically revised or a loose screw replaced, and two patients suffered from implant sequestration. Removal of the implants was indicated because the patients also showed mild (n = 2), moderate (n = 1), or severe lameness (n = 1), or the limb was swollen in the area of osteosynthesis (n = 1). The explants were colonized by Staphylococcus intermedius, in two cases with methicillin resistance. One dog had a pseudomonal and enterococcal infection.

### 3.16. Bacteriological Test Results

In 18 cases, swab samples were used to detect the pathogen, while in 53 cases, direct smears were prepared and analyzed. Seventy-one bacteriological examinations were analyzed in the Institute of Microbiology and Animal Diseases of the Freie Universität Berlin (Berlin, Germany). From these, 35 samples (49.3%) contained microorganisms. This affected 2 feline (n = 14) and 33 canine implants (N= 57). Six of these thirty-five samples contained more than one type of pathogen. Fungal pathogens were discovered in two implants, one with Aspergillus ssp. and the other with Candida ssp. In both cases, it was a mixed infection with bacterial germs and in both cases, dogs were involved. The isolated bacteria (42 isolates from 35 samples) were listed and among the 2 feline and 40 canine isolates, 27 included Staphylococcus ssp. and 5 Bacillus ssp. as the most common species (Table 6).

## 4. Discussion

Post-operative infections still count, even with the recent development of antibiotics and peri-operative systemic antibiotics, as the most common surgical complications and common findings are pain, delayed healing, osteomyelitis, implant loosening, and loss of implant function [3,4,5,9]. With the microbiological techniques used in the present study, microorganisms were detected in 35 out of 71 (49.3%) explants. It is interesting that fracture healing occurred without any noticeable complications in 26 animals, although pathogens were detected in some of these cases (n = 12, 46.2%). In contrast to this, in 42 patients with complications, microorganisms were only detected in 21 (50%) animals at the time of explantation. Because of these results, the clinical relevance of contaminated implants remains controversial, as some of the patients in this study were healthy and fracture healing was unremarkable despite the detection of pathogens on the implant. The results of our own study also show that only certain microorganisms, under certain conditions, show pathogenicity and are capable of causing a symptomatic infection. The spectrum of pathogens in the present study corresponds to those in previous studies [2,10,11,12,13,14], and the most frequently detected bacteria were Staphylococcus ssp. and especially those of the intermedius species. Factors like age and body weight appear to increase the risk of infection. These results were also confirmed by observations from Bardet et al. [15] and Brown et al. [16] but stand in contrast with other studies where hematogenous osteomyelitis was found in younger animals [14,17]. Ethridge et al. [18] demonstrated that wound infections are more common in intact male dogs than in neutered and intact female dogs, which is also differs from the results found in this study, where the sex was not correlated with an increased risk of developing an infection. A significant correlation between infection and the type of implant was not revealed. Neither the dynamic compression plates (DCPs) nor the locking plates were statistically significant factors in the development of a bacterial infection. However, the locking plates tended to appear to have a higher risk of infection than the DCP implants. This result does not agree with other studies, which found more frequent contamination in DCPs [19,20].

One explanation for this discrepancy could be the dimension of the implant (thickness, width) which significantly influences the risk of infection. Thinner 2.0 or 2.7 mm plates had a lower risk of infection (38.5%) than thicker 3.5 mm plates (62.5%).

Furthermore, our data were able to show that post-operative clinically noticeable impaired wound healing was associated with an infected implant in 87.5% of cases. Whether this extraordinarily high infection rate is the result of an iatrogenically caused implant infection or a coincidental finding remains unknown. The same applies to reosteosynthesis, where 66.7% of explants were infected after repeated procedures. This high infection rate is also likely to be caused by local soft tissue reactions from the initial intervention, as scar tissue with insufficient blood supply might promote infection in any reosteosynthesis procedure with metallic foreign bodies.

No comparative literature could be found relating a surgeon´s experience to the probability of an explant infection. In total, the orthopedic procedures were performed by eight surgeons. Two surgeons had the greatest expertise in such procedures and performed more than half (50/71) of the operations. No statistically significant association was found with regard to the risk of infection and the surgeon´s experience; however, numerically the infection rate appears to be higher among the experienced than among the inexperienced surgeons. The reason for this finding could be that the highly qualified surgeons operated on more complicated procedures such as open or comminuted fractures, as well as joint fractures or arthrodeses. Also, once the crucial phases of osteosynthesis (reduction, stabilization, first proximal and distal screws) were completed, further procedures and wound closure were often handed over to a less experienced surgeon. This could lead to prolonged procedure times. The literature shows that the risk of infection doubles every 70 min [21]. This is also consistent with human literature, where surgical site infections tend to increase with the duration of the procedure [22]. In the present study, and according to Bahn [23], it tended to decrease. Knobloch [24] found that neither the duration of the surgical procedure nor the individual surgeon was associated with a significant surgical risk for infection.

One person assisting led to a 42.9% risk of infection, whereas with two, it was 66.7%. Fourteen out of sixteen operations with three assistants attending were performed by an experienced surgeon. The risk of infection increased with the number of people attending. According to Carlson [25], the number of people involved in the operation, whether as assistants or just as spectators (students), is a risk factor for developing an infection. Every additional person in the operating room increases this risk of wound infection by up to 30% [21,25]. In our clinic, the standard team for every operation includes at least five people: one surgeon, two assistants (one of whom is a student), one surgical nurse, and one anesthetist. In addition, two students are allowed to observe each procedure. We believe that in a small animal teaching hospital such as ours, more experienced surgeons typically have a greater number of assistants or observers, primarily because of the complexity of the procedures they undertake. Despite this statement, a recent study in human medicine shows that the complication rate associated with having observers in a surgery is comparable to the reported data in surgeries with no observers, with no increase in the rate of infection or other post-operative and intraoperative complications [26]. Therefore, correlation between the number of people present in the operating room and the risk of implant infection remains controversial.

Even though the number of cases is low, it is interesting that out of a total of 35 infected explants, 21 (60%) were located in the forelimbs and 14 (40%) in the hindlimbs. The infected explants in the forelimb included radius/ulna fractures (10/16) and carpal joint arthrodeses (7/10). Reasons for the high rate of explant infections occurring especially in radius/ulna fractures are currently unknown, but one explanation could be the relatively thin soft tissue layer in the distal third of the forearm, in which approximately 42.8% of forearm fractures are located [27]. This area might have reduced vascularity, which could lead to a faster contamination. Factors facilitating the infection in carpal joint arthrodesis might include the reduced soft tissue cover, longer operation times with destruction of the articular surfaces, autologous cancellous bone transplantation, and the implantation of more than one plate [28]. These reasons for infected explants of the carpal arthrodesis are also likely to be applied to tarsal joints (4/6 infected).

Thirty patients were continued on antibiotic treatment after surgery in addition to the peri-operative administration. However, in 57% (n = 16) of those cases, the explant infection could not be prevented. On the other hand, 44% of explants (n = 16) were infected after only receiving one peri-operative administration of an antibiotic. Neither the antibiotic regimen, any additional injuries, the length of time between accidents and surgical treatment, the hospitalization time, nor any previous procedures significantly increased the risk of infection. Interestingly, 6/8 patients with a wound infection were hospitalized for at least 1 day. This observation between the increased risk of infection and longer hospitalization times was also confirmed by previous studies from human [29] and animal patients [21].

Implant contamination cannot always be radiologically detected using diagnostic imaging. Early radiographic signs, with almost 63% sensitivity and 57% specificity, are post-operatively widened soft tissue radioopacities and persistent gas inclusions [30,31]. Approximately one week post-infection, subtle peri-osteal proliferations (later noticeable due to bone resorption), osteolysis, sequestration, and blurred whitening zones around the implants may be noticed; however, imaging diagnosis of these infections is challenging because of several overlaps with non-infectious etiologies [32]. These findings were observed without clinical significance in 47% (32/68) of operations. Explants with obvious fracture healing disorders, screw loosening, or implant breakage were infected in 60% of cases. This applies even more to 73% of patients in whom there were functional disorders of the musculoskeletal system during the healing process and/or a misaligned healed limb. Dvořák et al. [33] already established that radiologically disturbed fracture healing may not necessarily be associated with clinical symptoms.

Bacterial osteomyelitis can be challenging to diagnose [17] and it was diagnosed in 9.8% of the dogs in the present study. This is in line with previous literature, where this complication was reported in 0.6% to 14.8% of cases [11,34,35]. Dogs had a mean age of 6.1 years, with a median of 7 years, and a mean and median weight of 35.8 kg and 39.9 kg, respectively. This is consistent with previous studies, where osteomyelitis was more common in dogs of medium and large breeds [15,36]. In this study, the most common radiographic findings were osteolysis (5/5), twice with sequestration and once with loss of bone substance. According to Walker et al. [37], radiographical changes are not always visible in osteomyelitis patients. The explants from affected dogs were most frequently infected by Staphylococcus intermedius, and this would be in line with studies [11,17,38,39] which showed staphylococci as some of the most common pathogens causing osteomyelitis. Laboratory advances including new molecular techniques now enable the detection of more fastidious microorganisms, such as Kingella kingae, which raises the need to reconsider the prevalence data of Staphylococcus aureus [14].

Limitations included the fact that the analyzed data were subject to the accuracy of the medical records and the sample size was only modest. These initial results should be a reason for further, much broader-based studies in which samples are collected during implantation but also when opening the surgical area and before wound closure and explantation, so that the issue of contamination or infection is followed up.

Currently, only a few clear recommendations for implant removal exist in human and veterinary medicine, such as implant-associated pain, implant failure, metal allergies, risk of peri-prosthetic fractures, functional limitations, or infection [40,41,42,43], and guidelines for implant removal are yet to be created [44]. In the present study, microorganisms were detected in almost 50% of the explants, but clinically relevant infection was only detected in five patients (7.3%), suggesting that the presence of a bacterial contamination does not necessarily appear to be a clear indication for implant removal, as long as clinical signs are not present. The incidence of bacterial colonization on metallic plate implants seems to be significantly higher than the incidence of osteomyelitis; the majority of animals are clinically healthy and bacteria are nevertheless present on the implants. However, this equilibrium can easily be impaired and a timely explantation of implants is recommended [45].

## 5. Conclusions

This document is the first in surgical small animal orthopedics in which risk factors for a possible explant infection were systematically investigated. Factors like body weight and age, location and type of plate, additional injuries like lung lesions, the surgeon’s experience, or the number of people present during the surgical procedure seem to influence the development of an infection. Of the animals, 60% showed osteolytic changes and 73.3% of those with dysfunctional mobility had an implant infection.

In the present study, microorganisms were detected in almost 50% of the explants, but a clinically relevant infection was only present in five patients (7.3%). This gives reason to continue studies questioning symbiosis, commensalism, eubiosis, and normal flora on explants or factors such as changed oxygen levels, pH values, temperatures, varied anatomical conditions, hygienic measures, or even antibiotic treatments that could affect the normal flora and thus promote the development of other pathogenic forms.

## Figures and Tables

**Figure 1 vetsci-11-00221-f001:**
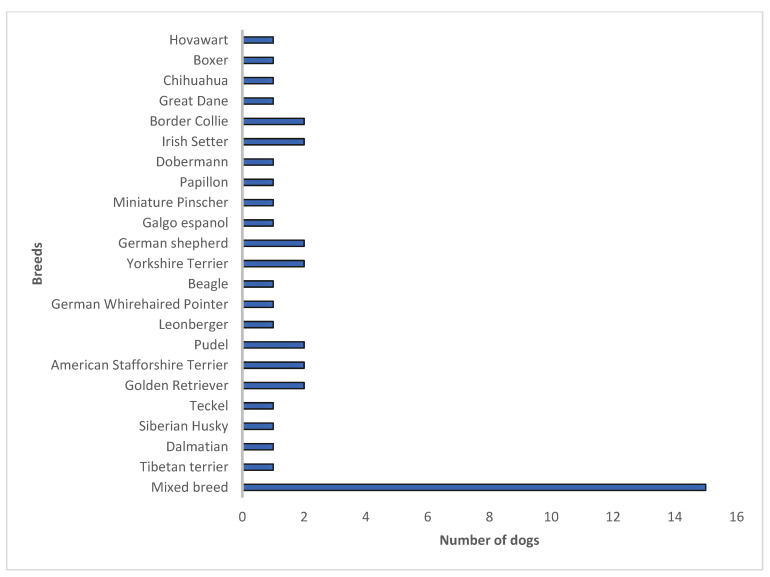
Dog breeds and amount in this study.

**Figure 2 vetsci-11-00221-f002:**
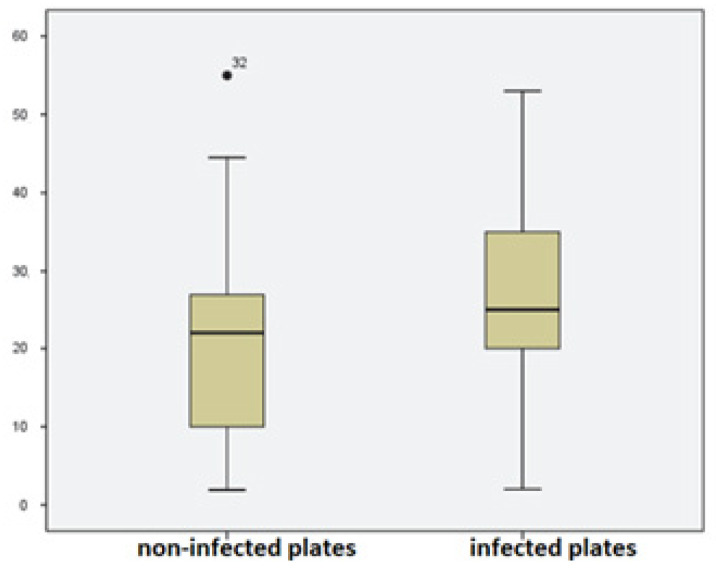
Microbiological findings and its relation to the dogs’ body weight.

**Figure 3 vetsci-11-00221-f003:**
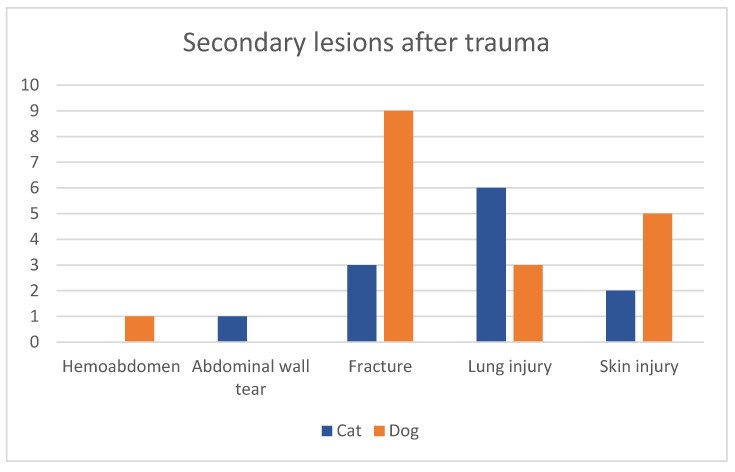
Number of secondary lesions in dogs and cats presented after trauma.

**Figure 4 vetsci-11-00221-f004:**
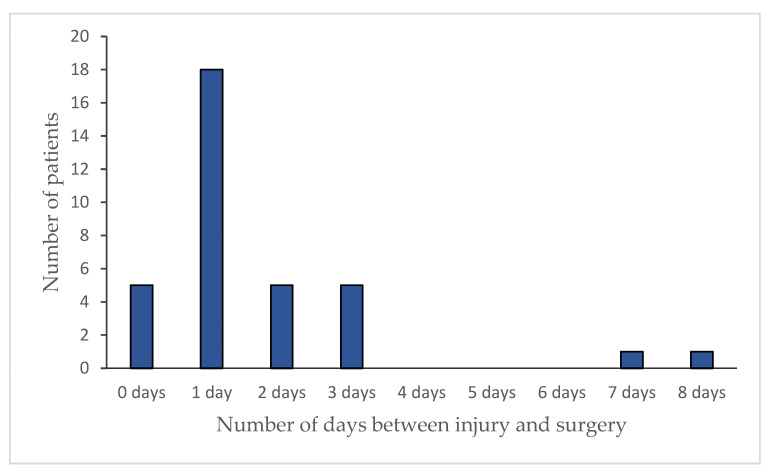
Number of patients and number of days between the diagnosis/occurrence of the lesion and surgical treatment.

**Table 1 vetsci-11-00221-t001:** Distribution of infected and non-infected explants in the different dogs’ body weight groups.

Weight in Kg	Explant	
	Infected	Non-Infected
0–5	4	4
5.1–10	2	2
10.1–20	6	5
20.1–30	9	11
30.1–40	1	6
40.1–50	1	2
>50	1	3
Total	24	33

**Table 2 vetsci-11-00221-t002:** Incidence of fracture location in canine and feline patients and distribution of infected and non-infected plates in the different affected body areas.

Fracture Localization		Dogs´ Incidence	Cats´ Incidence	Explants	
				Non-Infected	Infected
**Forelimb**	Articulatio carpi	6	4	3	7
	Radius	1	1	2	
	Ulna		1	1	
	Radius-ulna	14	2	6	10
	Articulatio cubiti	2		1	1
	Humerus	2	2	3	1
	Articulatio humeri	2		1	1
**Hindlimb**	Articulatio tarsi	5	1	2	4
	Tibia	7	1	3	5
	Tibia-fibula	2	2	3	1
	Articulatio genus	2		1	2
	Os femoris	7	5	11	1
	Os ilium	1			1

**Table 3 vetsci-11-00221-t003:** Total amount of extracted plates and their respective number of screw holes.

Number of Screw Holes	Number of Locking Plates	Number of Dynamic Compression Plates
5	0	1
6	6	2
7	0	2
8	9	11
9	0	1
10	9	4
12	7	4
13	0	1
14	3	2
16	3	0

**Table 4 vetsci-11-00221-t004:** Plate thickness and amount of explanted locking plates and DCPs.

Plate Thickness	Number of Locking Plates	Number of Dynamic Compression Plates
2 mm	9	9
2.7 mm	4	11
3.5 mm	15	7
4.5 mm	9	1

**Table 5 vetsci-11-00221-t005:** Days of hospitalization in infected and non-infected explants.

Hospitalization in Days	Explants	
	Non-Infected	Infected
0	13	13
1	4	6
2	6	6
3	7	2
4	2	3
5	1	2
6	1	1
8	2	0
13	0	2
Total	36	35

**Table 6 vetsci-11-00221-t006:** Incidence of bacterial species found on the infected explants of dogs and cats.

Bacterial Species	Incidence		N
	Dogs	Cats	
*Staphylococcus* ssp.	26	1	27
*Bacillus* ssp.	5		5
*Pseudomonas* ssp.	2		2
*Proteus* ssp.	1		1
*Enterococcus* ssp.	1		1
*Providencia* ssp.	1		1
*Enterobacteriaceae* ssp.		1	1
*Streptococcus* ssp.	1		1
*Arthrobacter* ssp.	1		1
*Micrococcus* ssp.	1		1
*Paenibacillaceae* ssp.	1		1

## Data Availability

Data is unavalaible.

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
