# Peer review of "Microbial Colonization of Explants after Osteosynthesis in Small Animals: Incidence and Influencing Factors"

_vetsci, 2024, doi:10.3390/vetsci11050221_

Round 1

Reviewer 1 Report

Comments and Suggestions for Authors

Authors should clarify or correct aspects of the paper regarding this reviewer comments:

L23.- Change "anti-biotics" for antibiotics

L 38.- In Keywords, change “infection ;hardware ;implants ;dogs ;cat” for infection; hardware; implants; dogs; cat

Authors should explain why they considered hardware as a keyword in this paper. 

L 40.-  In “.Introduction” please remove the period sign before the word Introduction

L 46.- Correct “post- operative” for post-operative”

L 62.- Please remove the period sign before the word Patients in “2.1. . Patients”

L115.- Please remove the period sign before the word Implants in “2.2. . Implants”.

L 146.- Please remove the period sign before the word Results in “3. . Results”.

L 220-221.- In table 2, check the space between words in Table legend “distribution of infected”.

L 404.- Please remove the period sign before the word Discussion in . Discussion”.

L 494.- Check the space between the word human and the reference number human[29] ”.

Author Response

Dear Reviewer,
Thanks a lot for your critical correction. All the suggested changes have been applied.

Best regards,

Reviewer 2 Report

Comments and Suggestions for Authors

Dear Sir/Madam.

Congratulations on a fine paper. I was curious could the authors draw a connection between higher infection rate with experienced surgeons and the number of people present in the operation room. In my experience there are more people in the operating room as observers when experienced surgeons operate. Another interesting factor was the time factor, maybe the authors could elaborate on this more, or provide further analysis?

Best wishes

Author Response

Dear Reviewer,

Thank you for your insightful feedback. We are pleased that you found the data intriguing and appreciate your suggestions.

As discussed in the manuscript, there remains debate regarding the correlation between infection rates and surgical operating times, as well as the presence of additional personnel in the operating room.

We propose that experienced surgeons may have more observers or students present due to the complexity of the procedures they perform. Additionally, our findings indicate that shorter procedures exhibited higher infection rates compared to longer ones, a phenomenon that warrants further investigation given the conflicting evidence in the literature.

To clarify this we added the following sentence in the discussion:
"We believe that in a small animal teaching hospital such as ours, more experienced surgeons typically have a greater number of assistants or observers, primarily because of the complexity of the procedures they undertake". This might be the explanation for higher surgical infections.

We believe that larger-scale studies with expanded sample sizes and comprehensive microbiological testing at various stages of the surgical process are necessary to elucidate the relationship between these factors and postoperative infections definitively. We believe that additional arguments may not be essential at this stage, especially considering the absence of significant new data to contribute.

Best regards,